# Circular Economy in Mountain Value Chains: The Case of Three PDO Cheeses

**DOI:** 10.3390/foods12213954

**Published:** 2023-10-29

**Authors:** Kamar Habli, Diana E. Dumitras, Emilia Schmitt, Isabella Maglietti Smith, Dominique Barjolle

**Affiliations:** 1Origin for Sustainability, Av. Parc de la Rouvraie 20, 1018 Lausanne, Switzerland; kamar.habli@origin-for-sustainability.org (K.H.); isabella@origin-for-sustainability.org (I.M.S.); 2Department of Economic Sciences, University of Agricultural Sciences and Veterinary Medicine Cluj-Napoca, 3-5 Manastur Street, 400372 Cluj-Napoca, Romania; 3Department of Agricultural Economics, Universidad de Cordoba, Campus de Rabanales, E-14071 Córdoba, Spain; sc2scsce@uco.es; 4Institute für Agrarwissenschaften, ETH Zürich, LFH B 4, Universitätstrasse 2, 8092 Zürich, Switzerland

**Keywords:** circular economy, cheese mountain products, PDO historical practices, 5Rs, code of practice, territorial development

## Abstract

The circular economy (CE) has shown promise for achieving several of the UN’s Sustainable Development Goals, replacing the linear system and reducing negative impacts on the environment. This research aims to assess the effective adoption of CE principles in three cheeses with geographical indication (GI) through an analysis of the practices identified in their respective value chains. Qualitative interviews show the persistence of historical practices that preserve the heritage behind the product, maintain autonomy in relation to external inputs and save energy or make intelligent use of by-products. Radical adoption of CE principles requires innovation to reduce the use of new inputs and greenhouse gas emissions. GI food products are generally not constrained by standards beyond those set by law, but their specifications can be modified, while respecting practices consistent with the link to the terroir. However, the remoteness of small businesses in deep rural areas, far from research centers, is slowing down the transfer of knowledge and the adoption of the latest technologies, particularly in mountainous areas. More participatory research and innovative initiatives are needed to ensure the transition to a circular economy for traditional mountain products, which are strongly linked to local culinary traditions and cultural identity.

## 1. Introduction

In recent years, several studies have been conducted to shape how CE can be adopted as a territorial approach to sustainability. The circular economy (CE) was introduced in 1970 as a model for sustainably transforming the linear economy. The linear system of production used in business models places high pressure on the environment in terms of continuously extracting new resources for production and exacerbating externalities like biodiversity and ecosystems losses, greenhouse gas emissions, pollution, and climate change in the process [1]. The associated negative environmental impacts, especially climate change and the rapid growth of the world’s population, are pushing policy makers and organizations to adopt a sustainable production model [2]. With the creation of the Ellen MacArthur Foundation, emphasis was placed on circular economy, specifically in political decision-making [1]. The Ellen MacArthur organization provides evidence-based research about the CE transition and its contribution to mitigate environmental challenges such as climate change [3].

In its report of the meeting on sustainable development, the United Nations [4] stated that CE holds particular promise for the achievement of several sustainable development goals (SDGs): SDG 7 on energy, SDG 8 on economic growth, SDG 11 on sustainable cities, SDG 12 on sustainable consumption and production, SDG 13 on climate change, SDG 14 on oceans and SDG 15 on life on land.

Circularity is conceptualized by the expression “resource-product-resource” [5], which highlights the new resource value that waste can hold in terms of environmental and economic valorization. It replaces the “end of life” of a product with a different concept found in waste management plans, such as reusing, recycling, and repurposing [6]. The waste hierarchy is easy to memorize thanks to the 5Rs approach: Refuse, Reduce, Reuse, Repurpose and Recycle (Figure 1). This approach is a way to reduce the leftovers of households or production systems [7]. Such waste management plans have been developed to introduce environmentally sustainable resolutions to the rising food waste issues and climate crisis. Thus, the 5Rs is destined to reduce waste generated and maximize resource efficiency at the production and consumption levels [8].

CE also promotes the moderate use of resources, thus reducing the waste and all environmental impacts of processes and increasing the use of renewable resources. It focuses on extending the life of products with material recycling, technological renovations and a return to the factory. Moreover, CE aims to increase the efficiency of the economic systems as a whole and to minimize negative externalities related to the release of toxic substances, soil, and pollution [1].

In 2020, the European Commission adopted a new Circular Economy Action Plan (CEAP), one of the major elements of the Green Deal agenda. It aims to contribute to sustainability and introduce the concept of circularity in the regions by promoting well-being, socio-economic growth, and the reduction of environmental pressure [9].

The most common definitions and action plans of circularity are the ones from ADEME (Environment and Energy Management Agency, Paris, France) and Ellen MacArthur (Isle of Wight, UK). However, in practice, the geographical and contextual dimension of the circular economy principles is still not clear enough [10].

CE is based on looping ecological practices and the circularity of materials that can be found in one’s own production line or from other companies. It promotes complementarities between different companies at local level, such as the exchange of machinery and materials between the actors of the same territory, an increase in job offers for the local community and the improvement of the quality of employees’ life by adopting circular practices (Recycling, Remanufacturing, Repair, Repurpose). This could provide more opportunities for the workers to find better jobs connected with the value chains (VC) [5]. CE promotes innovation as well, through the development of new technologies aimed at reducing the environmental impact [2].

Moreover, CE is a multi-level concept, where resource circularity is distributed at two complementary separate economic structures. “High-level” circularity includes R strategies adopted by cluster industries, whereas “lower levels” are targeted towards actors that engage towards circularity, but on a smaller scale. The combination of CE activities at lower levels provides a higher level of circularity [6].

Local production systems present regularities from one place to another. Among the characteristics of these systems, there is always geographical proximity between production units and a strong dependency on local resources, high-level sharing of know-how and active exchanges between the different actors. That is why the direct local business-ecosystem likely directly impacts the potential for CE practices. In the food sector, local production systems are in rural areas but the vast majority are built around and strongly connected with cities where their consumers are located. The more remote areas and/or larger-scale production systems that focus on one product are more oriented towards conventional markets and exports. It is thus adequate/logical to consider that such production systems might implement CE at lower levels.

One type of such food system where CE is adopted at lower levels can consist of geographical indications (GI). GIs are intellectual property rights and quality schemes that are used to distinguish products, mainly from the food sector, characterized by a specific quality and reputation linked to their geographical origin [11]. These products come from rural regions, mostly in Europe, where the specific product is an emblematic and often major production outlet in terms of mass, occupation, and impacts. They are also attached to traditions and artisanal productions produced in multiple, scarce, and remote farms in very rural and/or mountainous regions, where partnerships with other businesses for the reuse of resources may be particularly challenging.

Moreover, research shows that the economic dimension is considered more frequently when discussing the sustainability of GI [12]. This is because the Protected Designation of Origin (PDO) label aims to protect the intellectual property and its added value for profitability, whereas the environmental dimension is less sought and poorly discussed in research [13].

One rare study [14] showed that the environmental performance is almost the same for GI products as for conventional non-certified products. Per hectare, certified products pollute less than conventional ones, thanks to the code of practice or organic production that limit the use of fertilizers. Among all the environmental indicators, certified products scored better for food miles, thanks to local raw materials and consumption. However, for carbon and water footprints, the scores are worse than the conventional products.

Very little literature that discusses the circular economy in mountain food products or cheese products has been created. Other research is focused on related subjects such as technology of cheese production and by-products transformation, consumers’ perception of mountain cheese products and sustainable agri-tourism/pastoral tourism in mountain livestock farming, etc.

The environmental benefits and impacts of GIs systems are thus still not fully understood. On the one hand, they may prove to have conserved CE practices in the sense that they rely on local resources and local connections, but on the other hand, they might be too remotely located to benefit from efficient technologies and exchanges with many production lines in the area. Therefore, the legal definition of GIs traditionally does not address the sustainability dimension, as GIs users primarily focus on intellectual property rights, mostly to enforce the protection of the original name against misuses. The global context is pressing all GIs to examine how to preserve more natural resources and adapt to climate change.

Analyzing GIs in connection with CE also sheds further light on the geographical and contextual differences with regard to how CE can or cannot be implemented. This is why we aim to assess the environmental sustainability of geographical indications through the identification of CE principles found in three European PDO cheese products localized in mountain regions. This paper will help identify transitional innovative and collaborative pathways towards the CE and to set up good practices, defined from the VCs, to guide the producers to implement such transitions.

The three main research questions raised are: (1) Do GIs adopt circular and sustainable practices? (2) Are these practices historical or innovative? (3) Can the Code of Practice (CoP) highlight sustainable and circular aspects, or should it remain flexible?

The findings of these questions will provide a primary understanding of how GIs appropriate CE principles in their VC and the regulatory framework that can promote and support sustainable practices. This research will have a practical implication for producers, by building a cheese VC flowchart that highlights CE and sustainability elements found in the three studied cheese mountain products.

## 2. Materials and Methods

This qualitative study analyzes the implementation of CE for three PDO cheese VCs localized in European mountains regions, selected from the project MOVING: Tête de Moine in Switzerland, Alto Molise in Italy and Serra de Estrela in Portugal (Figure 1). As part of the European Horizon 2020, MOVING aims to create a network of VCs located in mountain areas in Europe. This network will co-develop policy frameworks, using a bottom-up approach, that contribute to the sustainability and resilience against common environmental challenges, particularly climate change [15].

In the EU regulation, mountain areas are defined in the legal definition No. 75/268 (1975) of Less Favored Areas (LFAs) [16]. A VC encompasses the entirety of processes, from the initial production stages to the final stage of consumption. PDO, as defined by the EU regulation No. 1151/2012 [17], are quality schemes for specific food and agricultural products that express a strong link to the geographical area where it is produced. All the process steps, except consumption, within the PDO VC must take place in the limited region specified in the CoP [18]. The CoP, known also as product specification, is an obligatory set of written rules that explains key characteristics of the product related to the name, the limited geographical area, the description of the production method, the link between the area, the organoleptic qualities, and the control bodies [19]. The objective of the EU GI quality labels is to add value by safeguarding the name of the product to highlight its distinctive characteristics linked to the traditional know-how and origin [18].

This analysis is based on evidence screening of CE principles among pre-existing qualitative data about the geographical and natural resources, the sustainability, and the resilience in their respective cheese VC. These data were collected by the VC experts of the project MOVING in 2022 and were used in the current research to contextualize each of the VCs. Two co-authors had a practical implication in the data collection phase in Switzerland. They were active in reviewing the synthesis reports based on the data collection for all MOVING case studies. They were also in close contact with the researchers of the other case studies, organizing 10 workshops with the partners of all MOVING case studies and several bilateral in-depth reviews of the cases.

We have selected three case studies which are part of the MOVING EU project. These cases are traditional cheeses, being European Protected Designation of Origin (PDO). These cases have been selected because cheeses are traditional in Europe, and because they are located in three different geographical mountain areas of Western Europe: in the Italian Apennines (caciocavallo Silano), in the Swiss Jura (Tête de Moine), both of which are located in “wet mountains”, and one in the Serra da Estrela for a PDO cheese of the same name in Portugal in Southern Europe (which are located in “dry mountains”). The mountains are typically far away from the research centers and consumers and suffer from harsh climate and steep terrain. For that reason, applying the CE principle can be even more challenging than for other value chains, and this makes it particularly interesting to explore the situation in cases exposed to severe conditions.

In the case of the “caciocavallo cheese”, the producers use different geographic denominations. In the Alto Molise region, where the MOVING project was implemented, different denominations are used, one being “caciocavallo Silano PDO” and another being “caciocavallo di Agnone” officially recognized (in Italy only) as “Prodotto Tradizionale Territoriale”.

Tête de Moine is a Swiss mountain type PDO raw cow-milk cheese produced in the Swiss Jura region, in particular in the Alps during summer season, and is managed by the interbranch organization known as “Interprofession”, as defined by Swiss law. It is a strong and democratic association composed of farmers, cheese makers and ripeners responsible for managing the quality, production, protecting the denomination from copy and imitations, as well as marketing of the product. This is all defined in the Ordinance (nr. 919.117.72) on the extension of mutual assistance measures of interbranch organizations and producer organizations.

Serra da Estrela is an artisanal PDO cheese made from sheep milk. It is produced from two local sheep breeds from the Cordilheira Central of Portugal. It is produced by several family businesses, in which men are responsible for the shepherding and women for cheese making [20].

Three experts on Tête de Moine, four experts on caciocavallo Silano and caciocavallo di Agnone and one expert on Serra da Estrela were invited to an interview during March and April 2023. Each expert collected available data about CE in his VC and answered the questionnaire administered via video conferencing. The three cheese VCs are different in terms of the institutional framework, the social and economic aspects as well as the natural and geographical context of each of the represented countries.

The interviewers introduced the transition from linear economy to CE and explained the waste hierarchy to guide the interview. The 8 experts presented their VC and explained all sustainable or circular practices that were employed in the past and innovative practices adopted recently according to the 5Rs (Figure 2). The questionnaire is organized into 3 main sections: Product Profile, Knowledge Assessment and 5Rs. Starting with a product profile questions (country, name, labels obtained, date of registration and international environmental standards), a knowledge assessment of the CE among the experts was completed using a scale from 1 (the weakest) to 5 (the strongest). The knowledge assessment included questions about their familiarity with the CE principles, their participation in CE projects, examples of applied CE practices in the VC, as well as their confidence in their knowledge related to this concept.

The 5Rs section is divided into 5 subsections related to one of the R practices. Each R is explained with a definition and a representative example. Four questions are repeated for each subsection, such as:Does the production part of your value chain include practices related to the “Refuse” of materials?Which of the mentioned practices in the previous question are traditional/historical?Which of the mentioned practices in the previous question are innovative?Are the practices mentioned above specified in the CoP?

In total, 8 experts participated in the interviews. Interviews were recorded, after asking for formal consent and taking the experts’ approval. Audiotapes were transcribed, serving as the primary data source. Videotapes were used to show the documents shared and determine nuances in the informants’ speech. In addition, interviewers took notes during the discussion to highlight the main elements relative to the study goals, especially circular aspects. The information was then structured in an interview report.

The qualitative data collected were analyzed according to two main criteria. Firstly, the practices identified were classified per VC based on the 5Rs categories. Secondly, each practice was also assigned to one of the following five categories: historical and still existing, historical abandoned, historical upgraded, innovative, and present in the CoP. This classification helped us to develop a visual representation of the CE practices identified within the VC. A flow chart of the cheese processing was developed to emphasize the CE practices identified. Moreover, an applicability analysis of possible circular practices to be applied in the cheese production sector was created as a guide of good practices for producers.

## 3. Results

From the interviews, it appears that CE is not very well known nor mastered by the experts. Based on the results, experts are averagely knowledgeable about CE (3.67 ± 0.52) and moderately confident in implementing its principles in their VC (3.67 ± 0.58). One VC out of the three participated in CE projects. Nevertheless, 22 practices contributing to the CE in cheese VCs have been identified and are presented and further analyzed in this chapter of the paper. A summarizing table of the practices can be found in the Table 1. 

### 3.1. Identified Practices Related to the 5Rs Framework

Several innovative or historical practices, specified in the code of practice or not, in line with the waste hierarchy are presented in Table 1 with the respective 5R category. Each of the practices can be integrated to one or several Rs. A total of 22 practices were identified among the three VCs. Certain practices are commonly seen in the cheese VCs, thus explaining the difference between the total of the practices and the total shown in Table 2. Practices related to the “Reuse” of materials in the cheese production, “Repurposing” the by-products and the “Refuse” of certain practices and inputs are repeated in the three PDO VCs. Four out of the twenty-two practices were mentioned in the CoP (18.2% of the total of the practices).

The results show that “Reduce” and “Repurpose’’ were the most widespread practices among these VCs (Table 2): a total of 11 and 8 practices, respectively, were reported. By-products are used for further transformation into other products or biogas, as soil fertilizer or as animal feed. These practices are not mentioned in the CoP but have existed for a long time as a necessity to be self-sufficient and to valorize the waste as much as possible.

With the industrialization and the legislative framework of the CoP, some of these actions have been abandoned or upgraded to better and more innovative alternatives. “Reduce” is associated with the use of green technologies to reduce emissions and the use of non-renewable energy.

“Reuse” and “Refuse” were moderately adopted: five practices were identified for each R), whereas “Recycle” was the least common, with only two occurrences. Practices related to the “Refuse” of certain materials exist in the VCs. Some of these practices are mentioned in the CoP. This is because the CoP strictly mentions the use of local pastures and breed, the materials and the machinery authorized. In the case of Serra da Estrela, a single press machine may be used by the producers.

However, results show that practices can be categorized in 6Rs categories instead of 5 (Table 2). 3 “Resale” practices were indeed additionally identified among the VCs that are not integrated in the waste hierarchy. This R includes products that leave the VC for other usage. At the farm level, lamb and wool are sold, in the case of Portugal. Calves, baled hay, and manure are also sold to dual-purpose local farms in Italy. Alto Molise cheese makers are most likely to pay firms to pick up the whey for other transformations. Otherwise, the whey is either dried or sold to pharmaceutical companies.

### 3.2. From Historical Practices to Circularity/Sustainability

Historical sustainable elements exist in all three products, including practices that contribute to the qualitative and traditional aspect of the VC, which are the focus of GIs.

Each of the practices identified can be classified into five main categories: “Historical, still existing”, “Historical abandoned”, “Historical upgraded”, “Innovative” circular practices and practices specified in the CoP or not (Table 3).

Historical practices were widely identified in all products. Only a few practices were “upgraded” or “abandoned”. While looking at the presence of these practices in the CoP, only historical and still existing practices are mentioned.

In the table in the Table 2, thirteen practices are listed under the category “historical” and derive from ancestral practices that have defined the product since its creation. The practices identified as historical are not only embedded in the local VC culture, but also contribute to CE, in different parts of the 5R framework.

Tête de Moine production includes more practices in terms of sustainability, such as the obligation for livestock to graze, the use of agricultural fodder and the maintenance of family farming on a human scale. The distance between the cheese factory and the farm is not allowed to exceed 20 km, and there are daily quality controls for raw milk. Milk is transported in less than 24 h and transformed into cheese, as a way of guaranteeing the high quality of the final product. On the other hand, Serra da Estrela cheese is an artisanal cheese that relies mainly on human activity and is produced from two local breeds. A certification system at the production level for each cheese by a casein mark and a registration of breed in gynecological books is implemented as a traceability system and quality guarantee. For Alto Molise, the main sustainable practice is dual-purpose breeding, and the main qualitative aspects are the animal farming practices and the production of local fodder.

In four cases, these practices are mentioned in the code of practices and are guaranteed by a strong legislative system. These elements highly depend on the history, the tradition, and the CoP of the product, as well as the juridical/governmental system of the country and the management system adopted by the producer himself.

Certain historical practices were gradually abandoned over time due to technological advancements. In the past, shepherds in Portugal made daily trips to the mountains. In summer, between July and August, sheep would go to the mountain areas to pasture and back to the pen where they slept and were milked. Currently, this practice is almost entirely abandoned by shepherds. Nowadays, farmers keep their sheepfold in mountain areas, and they are more concentrated in the foothills. Cattle used to produce Alto Molise cheese were left to graze in summer in the mountains of the Apennines. Over time, this 2Rs practice “Refuse, reduce” was also replaced with semi-intensive farming, leading to a decline in direct mountain pasturing and increase in stall feeding.

Certain producers have regained interest in some of these practices and, recently, started to reintroduce them to their VC. Dual-purpose breeding in Alto Molise VC was a very common sustainable practice in the past. Mixed milk and meat farming has three circular purposes: Refuse, Reduce and Repurpose, because having animals that produce both meat and milk means keeping less animals, reducing the need for feed and repurposing what was once only a waste or side-product (known as the “lower value” milk/meat). This practice was widespread and had a positive impact on biodiversity and maintaining habitats. The reduction in grazing along with the population and farms has led to a decrease in grassland and the development of forests. After a serious decline and abandonment of this practice, farmers are regaining interest in this type of breeding system.

In certain cases, by-products leave the VC for other transformations or can be transformed on site. One example is the use of dried whey in the pharmaceutical industry. In this regard, some cheese VCs are left with huge amounts of whey because they are unable to afford the extra costs of transporting or transforming it. As seen in Alto Molise VC, producers tend to pay firms from the same area to pick up the whey. Tête de Moine producers concentrate the whey to reduce the volume and transport larger amounts for other industries, whereas for Serra da Estrela producers, transforming the whey on site into a ricotta-like cheese called “Requeijão” is an important part of their revenue. This “secondary” cheese is also a PDO product.

On the other hand, many VCs are adopting technologies to reduce the consumption of non-renewable energy by installing solar panels and wind turbines or even digesting some by-products into biogas. It is essential to highlight that none of these innovative practices are covered by the CoP.

Recently, certain producers of Tête de Moine have been investing in a digestive plant that turns whey into biogas to valorize the huge quantities produced and reduce the consumption of non-renewable resources. This was achieved by the collective system put in place in the Swiss Jura, the “Interprofession” mentioned above, to support the producers in achieving economic diversification and sustainable growth.

For these VCs, the CoP specifies generally historical practices such as the origin of the inputs, the local breed and the machinery and materials used. They are strictly specified to preserve the tradition behind the product and to protect the factors that play a huge role in the characteristics of the product. For example, raw milk in Tête de Moine affects the taste and quality of the final products.

### 3.3. Applicability Analysis of Possible CE Practices along the Cheese VC: Guide of Good Practices

To better organize and structure the process, it is possible to create connections and collaborations between different stages of the VC. This represents the flux of materials and energy in a circular manner. Figure 3 is a standard flowchart for cheese production, presenting circular practices identified for each step based on the insights gathered from interviews. The practices are categorized using various colors as shown below.

As seen on the right, at the farm level, in the case of Serra da Estrela, the shepherds sell the wool to produce “burel” fabric. The lamb is as well sold and marketed as a PDO product from the region. In the Italian case, calves, baled hay and sometimes whey are sold for dual-purpose farms. This shows the connection of the cheese production with other types of VCs.

Certain PDO specifications limit the farmers to the use of the local pastures only, as seen in “Refuse” practices. Other quality products are allowed to be produced only with raw milk (“Refuse” of heat-treated milk), to market a cheese product with distinct organoleptic characteristics.

At the transportation level, producers tend to use reusable materials to move the milk from the farm to the cheese dairies, such as inox tanks or high-density plastic containers. Regardless of the transportation method, cheese VC can be geographically limited with certain PDO specifications linked to the local resources and the distance between the farm and cheese maker. For Tête de Moine, the distance rarely exceeds the kilometers specified due to the limited operational territory in the CoP and past historical proximity between the cheese factory and the farms. As explained by [23] the organoleptic characteristics of the product are not only correlated to the geographical proximity but linked to internal and external attributes. The geological and pedagogical specificities of the territory, as well as the social and cultural factors linked to the local community, are responsible for preserving the cultural and traditional identity of the product. For these reasons, the distance between the two production units is limited.

At the heart of the flow chart, numerous fluxes connected to the same production plant or outside the VC of materials and inputs are drawn (as shown in Figure 3). In the past, the pigsty was closely localized to cheese makers, assuring the distribution of the whey produced at the transformation level as feed for pork. However, this proximity has since been abandoned and is now rarely seen. Nowadays, whey is either dried and sold for pharmaceutical companies or transformed on site as a different type of cheese.

Certain by-products can be valorized and reused during cheese production to help close the loop of energy and materials. As such, biogas production on site can be used to generate renewable energy for the functioning of the transformation plant. This requires an important financial investment that can be funded by a group of producers, as is the case for Tête de Moine, for which producers are reusing whey to generate energy. Manure is still used as soil fertilizer in certain farms. In other cases, it is used to produce biogas.

The reuse of materials is a common practice among the cheese VC. Mold, wood boards, and plastic rings are reused several times over the years. In certain cheese processing, the Saumur (salted water bath) is reused multiple times before another one is prepared.

Since water is an essential component of cheese production, water collection or neutralization systems are put in place to reduce the usage of water and reuse it in the industry.

At the maturation stage, natural cellars can be specified in the CoP of PDO products. These cellars can maintain the necessary temperatures and humidity with no need for electric supply.

At the last two steps of the cheese VC flow chart, packaging was less frequently sought during the research. This is because certain VC are still testing new packaging to conserve quality and reduce plastic consumption. On site, Serra da Estrela cheese is sold with paper packaging. Recyclable or returnable packages can be an investment that will allow cheese producers to reduce single-use plastic and increase reuse and recycling.

Finally, before the cheese is distributed for consumption and sale, different quality controls are performed. As seen at the left of the chart, if the quality control of Tête de Moine shows that the cheese is not suitable for human consumption, it is often “Repurposed” as animal feed or used to produce fondue cheese, grated cheese or destroyed in the case of bad-quality cheese, whereas if Serra da Estrela cheese is not sold, it is re-integrated in the VC, matured for longer and marketed as an aged PDO version of the cheese known as Queijo Serra da Estrela Velho.

### 3.4. Evaluating Cost–Benefit of Circular Economy Practices

Based on the results from the interviews, an analysis of circular practices in the cheese VC was elaborated (Table 4). It provides some CE principles that can be put in place. These practices, historical or innovative, aim to act on five environmental targets: Energy, Water, Inputs and Materials, By-products and Farm and Industry Management.

These targets, as the most frequently emerging action plans identified during the interviews, are defined based on the environmental sub-themes found in SAFA guidelines by the FAO: Sustainability Assessment of Food and Agriculture systems (2014) [24].

The cost, expressed by an ascending 5 scale “+” sign, was estimated based on the discussions from the three interviews with the experts about the cost-efficiency of the practices or technologies. The benefits were also estimated and summarized based on the information collected during the interviews that took place with the experts of the VC.

Table 4 can act as a guide that will help producers to rethink their production system and guide them in implementing possible circular practices. Such evidence-based guidance represents successful CE practices or technologies adopted, which are aimed at possible environmental targets found in the cheese production system. This also helps producers understand the short- and long-term benefits, economically and environmentally, of implementing these practices in their production system.

### 3.5. Multidimensional Perspective of the Adoption of CE Principles in the GI Cheese VCs

The results from the interviews conducted shed light on different factors that seemed fundamental for the adoption of CE. Figure 4 is a scheme that represents the multidimensional elements for a successful adoption of CE principles in GI cheese VCs localized in mountain regions.

The natural capital encompasses the environmental characteristics, the local resources and ecosystems present in each of the regions. This includes elements such as the water, soil, biodiversity, the weather (rainfall, sunlight) and local breeds and animals.

The human capital represents the local knowledge and know-how, expressed through the technical and technological skills in utilizing the natural resources and benefiting from the geographical characteristics. These skills are demonstrated by the farm management techniques and heritage preservation, as evidenced by the Alto Molise region with its dual-purpose breeding. Local actors contribute significantly to cultural identity by preserving traditional and historical gastronomy rooted in the community, particularly within the CoP associated with certain GI products. For instance, Serra da Estrela cheese production is artisanal, with only one machine permitted within the CoP.

However, stakeholders lack access to research institutions. To address this gap, action should be taken to ensure knowledge is transferred, including insights into market trends. Research centers provide up-to-date studies and technologies, offering participatory education and peer learning that can empower the local human capital with innovative techniques aligned with CE objectives. Knowledge transfer covers not only the theoretical principles but also technical and practical support. This bidirectional exchange, facilitated through communication platforms, will remove the gap between the remoteness of mountain producers and research establishments, with a specific focus on CE principles.

Public actors and local authorities provide the constitutional and structural framework for transitioning towards a CE. Previous work on agriculture research impact [25] shows that if a constitutional and structural framework for agriculture research aims at reaching out towards the local actors, the change and adoption of new approaches and technologies increases. This was also suggested by the evidence gathered among all cases studies that most of the respondents were interested in receiving support from research and outreach to become more informed about the circular economy principles and related technologies. By providing financial resources and necessary infrastructure (water supply, transportation, power, sewers, electrical grids, telecommunications) in these mountain regions, CE projects can successfully shift the transition from a linear economy to sustainability. Additionally, these authorities can offer advisory services and grant access to innovation processes and research, the producers in shifting to sustainable practices and CE technologies.

Through organized communication between the three actors of the territory, emphasis is placed on the social and natural capitals of the region. Stakeholders, research institutions and local authorities can collaboratively adopt CE principles, facilitating a smooth transition from the linear economy. These intra-dimensions should consolidate their efforts, corresponding to the cycle shown in Figure 4, potentially leading to the enhancement of the VC by integrating CE principles.

## 4. Discussion

An economic system contributes to environmental sustainability when its activities help to preserve ecosystems and natural resources. This implies a transition towards renewable resources, waste-management plans and sustainable technologies. CE promotes ecological activities linked to minimizing waste, extending the life of resources and increasing efficiency. It is a “multi-level system” [6] that functions in a closed loop of energy and material flow. However, connecting CE to existing business is challenging, as it requires collaboration between all the actors, drastic changes in practices and perspectives at the production and consumption levels [2].

Local production systems such as GIs are characterized by the geographical proximity between production units, local resources, and the knowledge transfer over the years. With the current rise of climate change, the loss of biodiversity and socio-economic problems, the CE approach has been raised as a solution to the challenges of the territory in terms of resilience, valorization and mobilization of regional government and collective actors [26].

This research addresses the concept of CE within PDO cheese products localized in mountain areas. Three research questions have been raised and discussed throughout this study.
(1)Do GIs adopt circular and sustainable practices?

Mountain agriculture provides ecosystem services that help conserve biodiversity and natural resources. It supports the economic development of mountain regions using local and quality materials. From a societal perspective, agriculture in these areas helps conserve the traditional and historical gastronomy of the community. For these reasons, many European or international mountain products are registered as geographical indications to protect the heritage and maintain their production [27].

However, the LFA areas are characterized by low input, high cost and decreased yield due to the socio-economic constraints and environmental challenges [16]. Despite receiving direct payments to maintain agriculture activities, these marginal areas are often abandoned by farmers for more profitable and productive lands [27].
(2)Are these practices historical or innovative?

On one hand, based on the results, local producers successfully maintained historical practices that have existed for several decades in order to remain autonomous and independent and preserve the legacy behind the product. The concept of circularity in these rural regions is integrated in their practices via the sustainable use of natural resources such as the Reuse and Repurpose of by-products and Reduction in waste. These practices have been shared throughout the generations. In areas with geographical handicaps, machinery and innovation can be challenging to introduce due to a lack of technical knowledge, funds, and geographical conditions. As such, the production system is highly dependent on physical power.

Compared to their national reference, the farms that produce these three PDO cheeses are remarkably smaller (Table 5) and located at higher altitude, except for Tete de Moine, for which farms are larger. The small-sized farms and cheese dairies induce a low availability of the manpower for the transfer of knowledge between research centers. Thus, this extends the time taken to adopt innovative practices and achieve the transition towards sustainability and CE. This is explained in [2], where such a transition is challenging to achieve without the necessary infrastructures and technologies.

On the other hand, technologies—particularly green innovation—can be adopted to increase efficiency and decrease the negative effects on the environment. Valorizing by-products into new transformations plays an important role in the profitability for the producers, financially and to achieve self-sufficiency. This is the case where by-products had higher potential and benefit in including innovation rather than supporting the historical action.

Similarly, cheese GI producers are experimenting with different green technologies to help them adapt their production to the current environmental challenges and market needs, especially climate change, without threatening the reputation of their product. Tête de Moine producers are trying to implement a “cold maturation” phase, which prevents a production surplus, in certain periods around the year.

This confirms that CE does not necessarily imply modernization, technologies, and machinery. Historical knowledge holds value in reproposing previous habits that benefit the natural environment and the profitability of the VC. CE can and should also consider the living beings involved in these production systems, like cows, sheep and pigs on farms. Nevertheless, the CoP rarely highlights sustainability or circularity in the specifications. Very few of the identified practices were significantly present in the CoP. This is due to the fact that this label was created as an economic tool [12] to create niche markets and guarantee premium price for the producers by protecting the heritage and locally transmitted knowledge. This is also confirmed by [33]. The examination of CoP in 8 PGI and PDO products shows that the environmental aspect is not the top priority of stakeholders while seeking GI registration. Very rare environmental considerations are seen in the CoP of the case studies selected, except the specification of production methods, the size of sheep flocks and animal welfare due to the rarity of the breed used [33]. Nevertheless, it can be said that GI products do not meet as high environmental standards as organic farming, for example.
(3)Can the Code of Practice (CoP) highlight sustainable and circular aspects, or should it remain flexible?

Therefore, with the emerging needs and challenges, the CoP of GIs can evolve in terms of the tradition, history, and the characteristics of the product. Ref. [34] discusses that in certain GI cases around the world, innovation played an important role in adopting technological developments that lead to financial improvements to all stakeholders and the territory by ensuring environmental protection. These practices implemented side by side with tradition can help shape a sustainable VC and final product. However, research investigating the possible synergies between sustainability, tradition, and innovation in GI products is still scarce.

Debate around amendments of the CoP to balance between innovation and tradition are rising [35]. Ref. [35] indicates that restricted possibilities within the CoP for producers to find innovative solutions for the environmental and socio-economic problems may endanger their viability, whereas unlimited flexibility of the specifications can weaken the product’s identity.

Since GI are not “static museums” [35], transitions towards sustainability and adaptability against rising internal and external challenges are possible. Cautious changes, which preserve the product’s identity and characteristics, can help the resilience of these VCs.

Results show that the interviewees are not entirely confident in their CE knowledge. The exposure to CE concepts and transition depends on whether the experts were invested in CE projects within their VC or participated in research in the territory. To this end, a continuous learning process through knowledge exchange between actors of the same territory can help GI producers find long-term solutions. More importantly, a specific focus should be placed on re-educating the producers about the benefits of their previous circular habits and to support them in maintaining these historical actions, including those which may have been abandoned by the VC. This can be achieved through collective organisms, national institutions or regional authorities which can offer a space for exchanging knowledge and expertise with the stakeholders involved. External perspectives from researchers, policymakers, public authorities and consumers are crucial to identify the societal demands and the transitions needed, to raise awareness and to find sustainable solutions for the production system in place. In France, producers form an association known as “defense and management organization” (“Organisme de Défense et de Gestion”, ODG in French) that represents their interests as one and is supported by the national institute of origin and quality (INAO). INAO is directly linked to the French Ministry of agriculture. Such bodies can offer opportunities for communication between the different internal and external actors of the territory in question.

Moreover, a range of responsibilities are reliant on market dynamics and consumer behavior. Producers should embrace the adopted practices that contribute to the CE while marketing the product on the international market, especially to local consumers. These practices encompass not only innovative techniques, but also historical and traditional methods. Such a marketing strategy, in response to the rising socio-economic and environmental demands, will help valorize the production system used in the VC and recognize the associated heritage and patrimony, particularly within the local context.

Strengthening the set of actors participating in the decision-making process of the territory plays an important role in the institutional transformation. It helps define strategies, guidelines and regional policies that improve their living conditions as a community. Synergies between the public and private stakeholders organize responsibilities and functions among the actors and guarantees control of the VCs in question. They should also provide the technical and financial support for stakeholders to perform experimental projects for modern circular practices. This will help contribute to territorial development through achieving resilience and sustainability of the VC and the area [36]. In this regard, the “coordinated efforts” of the local actors and public institutions are fundamental for a long-term result towards a sustainable and circular territory.

## 5. Conclusions

The evidence screening of circular economy principles is captured in the three cheese case studies, but these findings are not conclusive for all GIs. The success of maintaining historical practices and implementing innovative practices of circularity is measured by identifying the 6Rs instead of 5Rs. This is because certain by-products leave the VC and are sold to another for ulterior transformation. Articles discussing the environmental sustainability of quality products through the identification of CE aspects are scarce. Additional research and a larger scope are required to identify the circularity aspects within GIs and local food VC.

By developing a cheese VC flow chart that stresses the CE practices and the sustainability components of the cheese mountain products, the current research provides theoretical and practical implications. Collaboration amongst VC actors is desirable to ensure the adoption of CE principles at all levels. Assuring the necessary foundation for a seamless cycle of multidimensional elements to adopt CE principles in less-favorable areas of cheese GI VCs is a crucial responsibility for policymakers. Future work could consider other categories of GI mountain products. Future research could consider other categories of GI mountain products. In the context of the MOVING project, our results will be discussed in foresight participatory workshops to take specific needs of the cheese VCs into consideration while reflecting on the policies that will ensure a better future for the mountain communities and ecosystem. The participation of local communities to tackle the challenges and opportunities that are based on the CE principles is key to addressing the Grands Challenges of the Climate Change.

Balancing the historical application with technological innovations can take place in local VCs without affecting the specifications and the identity of the product, especially in the case of the CoP of GIs. Validating the circularity of historical abandoned practices by re-educating the producers is necessary to erase the misinterpretation that CE implies. Circular and sustainable practices can include practices that do not include big investments and technological innovations. This can help the producers to achieve a sustainable and resilient production system, with the rise of environmental challenges and the aggravating socio-economic obstacles.

Considering that these products are localized in less favored mountain areas, a series of factors may alter the integration of new models of circular economy in their VCs. Production at small-scale sites in mountain regions is highly dependent on the physical activity and know-how of the community. This is also conveyed by a legislative framework of PDO products, with a specific code of practice to be met. Mountain producers may have limited access to the knowledge transfer between stakeholders and research institutes. Moreover, certain technological innovations are not adapted to operate in high-altitude regions with difficult infrastructure conditions. Financial and technical support for small local producers to put in place green technologies are still scarce.

This requires institutional support and participatory research at the local and territorial levels, especially in mountain regions, to accompany stakeholders in adopting circular economy practices that contribute to the preservation of the environment, the profitability of their VC and the conservation of the culinary identity of the territory. This calls for access to advisory and training to communicate up-to-date research, trends and innovation that may help local and GI producers mitigate the rising obstacles and implement adapted technologies to their needs and production. Cooperation and effective communication between the producers and different actors contribute to the strength of the territory and the elaboration of long-term solutions.

## Figures and Tables

**Figure 1 foods-12-03954-f001:**
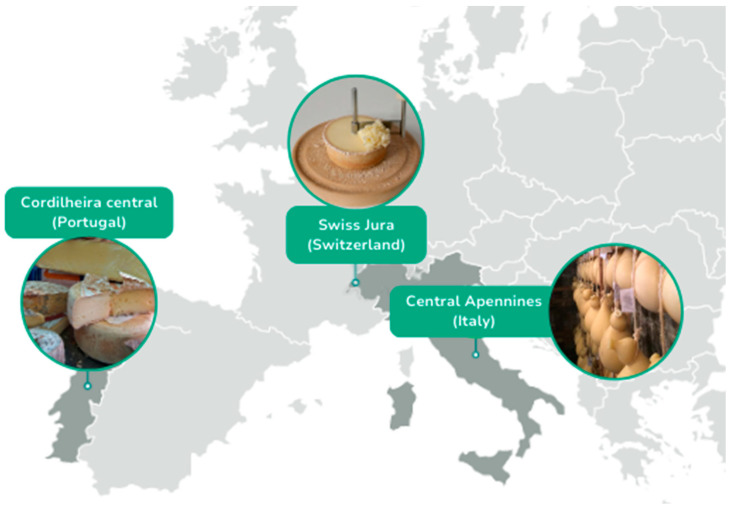
Map showing the location of the three cheese products.

**Figure 2 foods-12-03954-f002:**
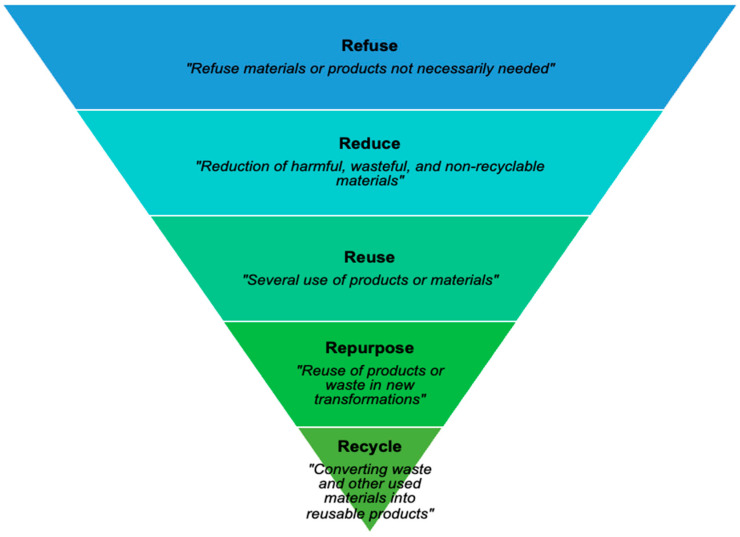
Diagram of the waste hierarchy (5Rs) adapted from WHO [21].

**Figure 3 foods-12-03954-f003:**
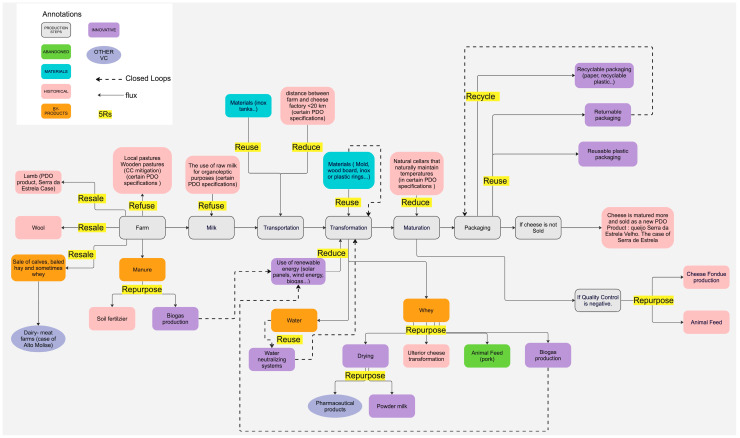
Flow chart of the cheese processing with the circular practices identified.

**Figure 4 foods-12-03954-f004:**
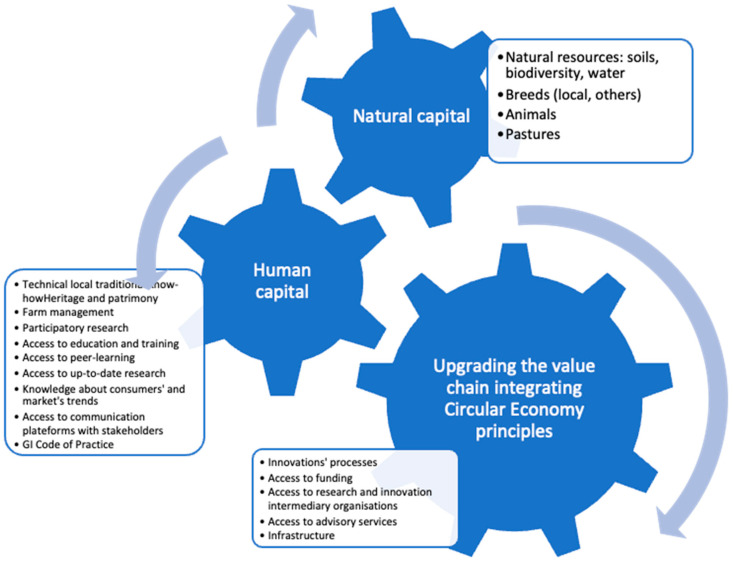
Cycle of multidimensional elements for adopting CE principles in less favored areas of GI cheese VCs.

**Table 1 foods-12-03954-t001:** Practices identified in the cheese VCs.

Practice	Purpose/Use	5Rs	Type of Practice	In the CoP or Not in the CoP	Number of VCs Using This Practice
**General practices**					
Wood for heating	Some cheese factories use wood for heating to reduce non-renewable energy consumption.	Reduce	Upgraded	Not in the CoP	1
Heat collection system	Some cheese factories are implementing heat collection systems to reduce energy consumption.	Reduce	Innovative	Not in the CoP	1
Local pastures	To produce certain PDO cheese, only native breeds, local pastures and hay are authorized. Imported feed for milking cows is refused.	Refuse	Historical	**In the CoP**	**3**
Reuse of manure	Manure is used as soil fertilizer.	Repurpose	Historical	Not in the CoP	3
	Manure is used to produce biogas.	Reuse	Innovative	Not in the CoP	1
Reuse of whey	Whey is used as animal feed.	Repurpose	Abandoned	Not in the CoP	2
	Whey is dried and used to produce powdered milk or pharmaceutical products.	Repurpose	Upgraded	Not in the CoP	2
	Whey is used to produce other Ricotta-like cheese: Requeijão in Portugal. This cheese is an important part of the revenues and is a PDO product.	Repurpose	Historical	Not in the CoP	1
	Whey is used to produce biogas. This is an investment made by collective actions.	Reuse	Innovative	Not in the CoP	1
Reuse of Wool	Wool is sold, usually by the farmer, to produce fabric.	Resale	Historical	Not in the CoP	1
Reuse of materials	Materials are reused several times such as mold, Saumur, wood board, inox or plastic rings…	Reuse	Historical	**In the CoP**	**3**
ISO (International Organisation for Standardization) 22000 certification [22]	The ISO 22000 for Food safety management refuses certain hygienic materials.	Refuse	Innovative	Not in the CoP	1
Use of renewable energy	Solar panels and wind energy to reduce consumption of energy.	Reduce	Innovative	Not in the CoP	1
Vending machine	Vending machines for cheese sale to reduce transportation.	Reduce	Innovative	Not in the CoP	1
Water reuse	Efficiency programs and water neutralizing systems in order to recycle and reuse water.	Reuse and recycle	Innovative	Not in the CoP	2
Reusable packaging	Some cheese producers are adopting reusable packaging instead of single-use.	Reuse	Innovative	Not in the CoP	3
Reduce plastic	Plastic packages are being replaced with laminated cardboard to reduce plastic use.	Reduce	Innovative	Not in the CoP	3
Recycling of packaging	Packages are sorted, cartons are recycled. Returnable containers at sale points and in shop supplies are possible.	Recycle	Innovative	Not in the CoP	3
**Specific practices**					
Limit number of cows per hectare	For Alto Molise, there is a limit to the number of cows per hectare to meet subsidies requirements in Italy.	Reduce	Historical (Legislative)	Not in the CoP	1
Resale of by products	In the case of Alto Molise, cheese makers pay local firms to pick up the whey. Calves, baled hay and manure can also be sold to farms or dairy-meat farms in the area. This is the case because Alto Molise producers do not always have the resources to transport the whey or transform it on site.	Resale	Upgraded	Not in the CoP	1
	In the case of Tête de Moine, whey is sold dried concentrated to reduce its water volume for transportation.	Repurpose, Resale	Innovative	Not in the CoP	1
Local inputs and one single machine	In the case of Serra da Estrala cheese, the local ingredients are important. In the CoP, only one machine can be used in the production. This is because the cheese is an artisanal historical product based on human activities.	Refuse	Historical	**In the CoP**	**1**
The use of Girolle	To reduce food waste and facilitate the consumption of Tête de Moine, Girolle was produced in order to sell the cheese as “Rosette”.	Reduce	Historical	**In the CoP**	**1**
The use of Rosomats	For the Tête de Moine VC, Rosomats are manufactured to reduce the amount of transportation and plastic packages.	Reduce	Innovative	Not in the CoP	1
Cold storage and maturation	Cold storage and slow maturation are being experienced in Tête de Moine to reduce over production and seasonal variation between milk production and consumption.	Reduce	Innovative	Not in the CoP	1
Dual purpose breeding	In Alto Molise VC, Mixed meat-dairy farming had a positive impact on biodiversity and maintaining habits	Refuse, Repurpose, Reduce	Abandoned	Not in the CoP	1
Mountain grazing	In the past, shepherds involved in Portuguese cheese-making made daily trips to mountains to the pasture and back to the pen, where the animals sleep and are milked.	Refuse, Reduce	Abandoned	Not in the CoP	1

**Table 2 foods-12-03954-t002:** Classification of the practices identified in the cheese VC based on the 6Rs waste hierarchy by VC.

Categories	Historical	Innovative	In the CoP	Total 6Rs/VC
**Tête de Moine**				
Refuse	0	1	1	2
Reduce	1	7	1	9
Reuse	2	4	1	7
Repurpose	3	1	0	4
Recycle	0	2	0	2
Resale	0	1	0	1
**Total category by VC**	**6**	**16**	**3**	**25**
**Serra Da Estrela**				
Refuse	0	0	2	2
Reduce	1	1		2
Reuse	1	2	1	4
Repurpose	3	0	0	3
Recycle	0	1	0	1
Resale	2	0	0	2
**Total category by VC**	**7**	**4**	**3**	**14**
**Alto Molise**				
Refuse	2	0	1	3
Reduce	2	1	0	3
Reuse	0	0	0	0
Repurpose	3	1	0	4
Recycle	0	0	0	0
Resale	2	0	2	4
**Total category by VC**	**9**	**2**	**3**	**14**
**Total per category**	**13**	**13**	**4**	**26**

**Table 3 foods-12-03954-t003:** Number of practices for each of the 5 categories identified in the cheese VCs.

Categories of Practices	Number of Practices
**Historical (in Total)**	**13**
Historical, still existing	7
Historical, abandoned	3
Historical, upgraded	3
**Innovative**	**13**
**Practices in the CoP**	**4**

**Table 4 foods-12-03954-t004:** Applicability analysis of possible circular economy practices in the cheese VC.

Target 1: Energy			
Practice	Circular Process	Cost	Benefits
Producing biogas from by-products	Reuse	++++	Increase sufficiency, reduce energy cost over time, valorizing by products.
Using renewable energy sources: water, solar or wind energy	Reuse, Reduce	+++++	Increase sufficiency, reduce energy cost over time, decrease the use of non-renewable energy and CO_2_ emissions.
CO_2_ collection system	Reuse, reduce	+++++	Reuse heat and CO_2_, decrease emissions, reduce energy use and cost.
**Target 2: Water**			
Water reuse and recycling: neutralizing systems	Recycle, reuse	+	Reduce water consumption, reduce dependency and increase efficiency.
Water collection system in cistern/tanks and reuse it for cleaning purposes	Reduce, reuse	++++	Reduce water consumption
**Target 3: Inputs and materials**			
Use of recyclable materials	Recycle	+	Conserves energy, reduces pollution, reduces GHG, and conserves natural resources.
Exclude single use plastic, in production and packaging	Refuse	+	Reduces pollution, saves resources.
Multiple use of materials such as containers and mold.	Reuse	+	Reduces pollution and waste, save resources, reduce GHG emissions
Encourage the use of local inputs: breed, pastures, resources…	Reduce, refuse	+	Reduce transportation costs, reduce GHG emissions, contribute to the region’s sustainability.
**Target 4: By-products**			
Producing other products from byproducts.	Repurpose	++	Increase profitability, create jobs, reduce waste, valorize byproducts.
Use of by products for other function	Repurpose and resale	++	Increase profitability, create jobs, reduce waste, valorize byproducts.
**Target 5: Farm and industry management**			
Efficiency increase	Reduce	+++	Reduce the number of inputs used to produce the same expected quantity, reduce losses and waste.

Symbols: +: Really cheap; ++: Cheap; +++: Average; ++++: Expensive; +++++: Really expensive.

**Table 5 foods-12-03954-t005:** Some characteristics of size of the farms that produce the 3 PDO cheeses and their national references.

	Average Farm Area per Hectare	Number of Producers	Number of Cheese Dairies	Tons of Cheese Produced per Year	Average National Size for the Milk-Producing Farms
**Alto Molise**	7.1 [28]	119 [20]	15 [20]	750 [28,29]	12 ha [30]
**Tête de Moine**	30 (mentioned by the expert)	245 (mentioned by the expert, 2022)	9 [20]	245 (mentioned by the expert, 2022)	21.6 ha 28.1 ha [31]
**Serra da Estrela**	4.6 (Seia, municipal level)	126 [20]	29 [20]	0.123 [20]	13.7 ha [32]

## Data Availability

Data collected for this paper are available in the public reports of the H2020 European MOVING project, complemented by available data collected for this paper by the authors.

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
