# Peer review of "Circular Economy in Mountain Value Chains: The Case of Three PDO Cheeses"

_foods, 2023, doi:10.3390/foods12213954_

Round 1

Reviewer 1 Report

This is a very interesting paper which analyses the circular economy in mountain value chains. The authors use the case of three quality cheeses to illustrate sustainable practices. While the research is innovative and timely, and well-developed, there are some issues the authors should revise. The authors should revise the introduction and include more recent references about the topic. These references should also include examples of previous cheese-based research in mountain areas, in food studies, which would improve the contextualisation and the implications of the paper. In the methods, the authors should explain why the three examples are important and how the experts were selected. The authors should also provide details about the questionnaire and the process of data analysis, for example, in relation to the identification of the categories. The authors should support the presentation of the cheeses with previous references. Also, the authors should improve the dialogue with previous research in the discussion (they should also use the references added in the introduction). In the conclusion, the authors should expand the theoretical and practical implications, limitations and opportunities for future research.

Language is ok.

Author Response

Dear Reviewer,

Thank you for your time to review our paper and your recommendations. All the suggestions were highly appreciated and helped us to improve the quality of the manuscript. Editing and language were checked in the entire manuscript. The changes are highlighted in yellow in the text.

Comments & Suggestions

Modification

Review Report 1

Revise the introduction & include more recent references

The references should also include examples of previous cheese-based research in mountain areas, in food studies, which would improve the contextualisation and the implications of the paper

Thank you for this comment.

We included more recent references in the introduction. Please refer to the article for the modifications and additions.

However, we did not find any suitable articles about cheese-based research in mountain regions or food studies that serve our article.

Line 38-44: The associated negative environmental impacts, especially climate change and the rapid growth of the world’s population are pushing policy makers and organizations to adopt a sustainable production model [2] . With the creation of the Ellen MacArthur Foundation, an emphasis was placed on circular economy, specifically in political decision-making [1]. Ellen MacArthur is an organization that provides evidence-based research about the CE transition and its contribution to mitigate environmental challenges such as climate change [3] ...

Line 52-53: It replaces the “end of life” of a product by different concept found in waste management plans, such as reusing, recycling and repurposing [6]. The waste hierarchy is easy to memorize thanks to the 5Rs approach: Refuse, Reduce, Reuse, Repurpose and Recycle (Figure 1). This approach is a way to reduce the leftovers of households or production systems [7]. Such waste management plans have been developed to introduce environmentally sustainable resolutions to the rising food waste issues and climate crisis. Thus, the 5Rs is destined to reduce waste generated and maximize resource efficiency at the production and consumption levels [8].

Line 80-87: CE promotes as well innovation, through the development of new technologies aimed at reducing the environmental impact [2].

Moreover, CE is a multi-level concept, where resource circularity is distributed at two complementary separate economic structures. “High-level” circularity includes R strategies adopted by cluster industries. Whereas, “lower levels” are targeted towards actors that engage towards circularity, but on a smaller scale. The combination of CE activities at lower levels provides a higher level of circularity [6].

In Methods: explain why the three examples are important and how the experts were selected.

From “Three experts of Tête de Moine, four experts of Alto Molise VC and one Serra da Estrela expert were asked for an interview during March and April 2023. The experts were selected from the MOVING project, since each expert did collect available data about CE in his VC and answer the questionnaire administered by video conferencing.”

To “We have selected three case studies, beeing part of the MOVING EU project. These cases are traditional cheeses, being European Protected Designation of Origin (PDO). These cases have been selected because cheeses are traditional in Europe, and because they are located in three different geographical mountain areas of the Western Europe: in the Italian Appenines (Caciocavallo Salina), in the Swiss Jura (Tête de Moine), being both in “wet mountains”, and one in the Serra da Estrela for a PDO cheese holding the same name in Portugal in the Southern Europe (being in a « dry mountains »). Mountains are typically far away from the research centers and consumers, and suffer from harsh climate and steep terrain. For that reason, applying CE principle can be even more challenging than for other value chains, and this is particularly interesting to explore the situation in cases exposed to severe conditions.  Three experts of Tête de Moine, four experts of caciocavallo Salina and di Agnone and one from cheese Serra da Estrela expert were asked for an interview during March and April 2023. Each expert did collect available data about CE in his VC and answered the questionnaire administered by video conferencing. The three cheese VCs are different in terms of the institutional framework, the social and economic aspects as well as the natural and geographical context of each of the represented countries. In the case of caciocavallo, the situation is complex, as in the region Alto Molise where the MOVING project was implemented, different cheeses are produced, one being caciocavallo Salina PDO, another being caciocavallo di Agnone recognised in Italy only as “Produtto Tradizionale Terrioriale”.

Provide details about the questionnaire and the process of data analysis, for example, in relation to the identification of the categories.

From” The interviewers introduced the transition from linear economy to CE and explained 155 the waste hierarchy to guide the interview. Experts presented their VC and explained all 156 sustainable or circular practices that were done in the past and innovative practices 157 adopted recently along the 5Rs (Figure 2). There was also a knowledge assessment of the 158 CE among the experts using a scale from 1, being the weakest, to 5, being the strongest. “

To” The questionnaire is organized into 3 main sections: Product Profile, Knowledge Assessment and 5Rs. Starting with a product profile questions (Country, name, labels obtained, date of registration and international environmental standards), a knowledge assessment of the CE among the experts was completed using a scale from 1, being the weakest, to 5, being the strongest. Knowledge assessment includes questions about their familiarization with the CE principles, their participation in CE projects, examples of applied CE practices in the VC as well as their confidence in their knowledge related to this concept.

The 5Rs section is divided into 5 sub-sections, related to one of the R practices. Each R is explained with a definition and a representative example. Four questions are repeated for each sub-section, such as:

•           Does the production part of your value chain include practices related to the refusal of materials?

•           Which of the mentioned practices in the previous question are tradition-al/historical?

•           Which of the mentioned practices in the previous question are innovative?

•           Are the practices mentioned above specified in the CoP? “

We added as well a sentence to explain the need for th category classification.

From: “...practice was as well assigned to one of the following five categories: historical and still existing, historical abandoned, historical upgraded, innovative, and if present in the CoP”

To “ as well assigned to one of the following five categories: historical and still existing, historical abandoned, historical upgraded, innovative, and if present in the CoP. This classification helps to develop a visual representation of the CE practices identified within the VC. A flow chart of the cheese processing was developed to emphasize the CE practices identified. Moreover, an applicability analysis of possible circular practices to be applied in the cheese production sector was elaborated as a guide of good practices for producers.

Support the presentation of the cheeses with previous references

Unfortunately, we did not find any suitable articles about cheese-based research on Circular Economy in mountain regions or food studies that serve our article.

improve the dialogue with previous research in the discussion (they should also use the references added in the introduction).

We included some more information and references from the introduction in the discussion. Please refer to the article (lines 521, 524, 562).

In the conclusion, the authors should expand the theoretical and practical implications, limitations and opportunities for future research.

Thank you for the suggestion. We added

Line 669-682 “By developing a cheese VC flow chart that stresses the CE practices and the sustainability components of the cheese mountain products, the current research provides theoretical and practical implications. Collaboration amongst VC actors is desirable to ensure the adoption of CE principles at all levels. Assuring the necessary foundation for a seamless cycle of multi-dimensional elements to adopt CE principles in less-favorable areas of cheese GI VCs is a crucial responsibility for policymakers. Future research could consider other categories of GI mountain products. As well, in the context of the MOVING project, our results will be discussed in foresight participatory workshops for taking specific needs of the cheese value chains into consideration while reflecting about the policies that will ensure a better future for the mountain communities and ecosystem. The participation of local communities to tackle the challenges and opportunities that are based on the circular economy principles is a key for addressing the Grands Challenges of the Climate Change.”

Reviewer 2 Report

This paper assesses the effective adoption of CE principles in 3 cheeses with geographical indication (GI) through an analysis of the practices identified in their respective value Chains. The methods used are correct and the writing logic is good. The data is also great, but the contributions and conclusions of the paper are not well. Here are some comments.

First, the authors do not state clearly their contributions to the literature. This should be added either in the introduction section or in the literature review section

Second, there are only three cheese products being analyzed in this paper. How is the representativeness of the sample?

Third, the conclusion section has to expand the discussion on the policy implications of the empirical findings.

Fourth, the paper only focuses on the Switzerland region, what are the meanings of the results for the other countries?

Minor editing of English language required

Author Response

Dear Reviewer,

Thank you for your time to review our paper and your recommendations. All the suggestions were highly appreciated and helped us to improve the quality of the manuscript. Editing and language were checked in the entire manuscript. The changes are highlighted in yellow in the text. 

Comments & Suggestions

Modification

Review Report 2

the authors do not state clearly their contributions to the literature. This should be added either in the introduction section or in the literature review section

Thank you for your valuable comment. Indeed, we should add our contribution.

Please refer to line 144, after the three research questions where we highlight the contribution of this article to the literature and we address the research gap.

Second, there are only three cheese products being analyzed in this paper. How is the representativeness of the sample?

This research was developed and carried out as in the framework of the EU project MOVING. This project groups 23 mountain regions found in 16 countries around Europe.

From these regions, only 12 products are geographical indications, from which 3 are cheese PDO/PGI mountain VCs.

Third, the conclusion section has to expand the discussion on the policy implications of the empirical findings.

Thank you for the suggestion. We added “By developing a cheese VC flow chart that stresses the CE practices and the sustainability components of the cheese mountain products, the current research provides theoretical and practical implications. Collaboration amongst VC actors is desirable to ensure the adoption of CE principles at all levels. Assuring the necessary foundation for a seamless cycle of multi-dimensional elements to adopt CE principles in less-favorable areas of cheese GI VCs is a crucial responsibility for policymakers. Future research could consider other categories of GI mountain products. As well, in the context of the MOVING project, our results will be discussed in foresight participatory workshops for taking specific needs of the cheese value chains into consideration while reflecting about the policies that will ensure a better future for the mountain communities and ecosystem. The participation of local communities to tackle the challenges and opportunities that are based on the circular economy principles is a key for addressing the Grands Challenges of the Climate Change.”

the paper only focuses on the Switzerland region, what are the meanings of the results for the other countries?

The paper represents mountain cheese value chains and presents examples from the three cases from Portugal, Switzerland and Italy, as seen in the Table 1 and in the results. As an example,

Line 310: 3 “Resale” practices were indeed additionally identified among the VCs that are not integrated in the waste hierarchy. This R includes products that leave the VC for other usage. At the farm level, lamb and wool are sold, in the case of Portugal. Calves, baled hay and manure are also sold for dual purpose local farms in Italy. Alto Molise cheese makers are most likely to pay firms to pick up the whey for other transformations. Otherwise, Whey is either dried and sold for pharmaceutical companies.

Line 326: Tête de Moine production includes more practices in terms of sustainability, such as the obligation for livestock to graze, the use of agricultural fodder and the maintenance of family farming on a human scale. The distance between the cheese factory and the farm is not allowed to exceed 20 km, and there are daily quality controls for raw milk. On the other hand, Serra da Estrela cheese is an artisanal cheese that relies mainly on human activity and is produced from two local breeds. A certification system at the production level for each cheese by a casein mark and a registration of breed in gynecological books are done as a traceability system and quality guaranteed. For Alto Molise, the main sustainable practice is dual-purpose breeding and the main qualitative aspects are the animal farming practices and the production of local fodder.

Reviewer 3 Report

This paper makes an important and original contribution to a topic of significance. It provides new insights that will be of interest to researchers, while also providing useful guidance for practitioners and their practice. The following comments are aimed at seeking clarification on points through the paper.

·      Line 28: 1970s, not 70s, for clarity

·      Line 33: Explain what is the Ellen MacArthur Foundation

·      Line 65: spell out VC

·      Line 75: in hypothesis: lower levels than what?  And if this hypothesis is going to be explored, needs to supported by more reference to the literature

·      Line 77: Fuller definition of GI needed

·      Line 112: Choose either ‘should’ or ‘how’ for the question

·      Line 113: After the 3 questions, add a sentence indicating how asking and answering these questions provides a basis for contributing to the literature

·      Line 135: Clarify that the data was collected by other researchers, if this is the case. And clarify the role of the authors of the paper in relation to data collection and analysis. Given that no quotes are presented from the interviews, please provide some more detail on how the interviews were analysed

·      Line 141: does this mean that the total number of participants for the paper is 8?  It is fine if yes, but need to clarify this is the case.

·      Line 143: Are all the participants male? ie use of word ‘his’

·      Line 146: define ‘interprofession’

·      Line 158: More specifics needed on what was involved in the knowledge assessment

·      Line 162: Consent is mentioned, any other ethics processes involved. Was formal ethics clearance obtained? Was permission obtained to name the organisations?

·      Line 168: how were the 5 categories of practice developed

·      Line 167: more clarity is needed here ie restate that this is based on 8 experts, if that is the case.

·      Line 192: The Table is good. However, with each line of the table, is it possible to identify how many of the 3 VCs were using the practice?

·      Line 351: Figure 3 is an important contribution

·      Line 360: A scale of 5 + for cost is mentioned. How was it decided where a particular practice fit in terms of the scale?  It is mentioned that they are estimated and summarised. What is the basis for the estimate?

·      Line 362: Another sentence is needed to explain how Table 4 will help producers rethink their systems. While the Table contains important information, it is not clear how it will be operationalised by producers

·      Line 393: It is claimed that public actors and local authorities emerge as the key actors. This might be the case, but what evidence is the claim based on?

·      Line 455: I would suggest not introducing other examples at this point.

·      Towards the end of the paper, return to the central questions and provide answers. Also indicate to what extent the hypothesis has been supported.

·      Overall, the paper requires a careful final proof read, for written expression

A careful final proof reading is required

Author Response

Dear Reviewer,

Thank you for your time to review our paper and your recommendations. All the suggestions were highly appreciated and helped us to improve the quality of the manuscript. Editing and language were checked in the entire manuscript. The changes are highlighted in yellow in the text.

Comments & Suggestions

Modification

Review Report 3

  Line 28: 1970s, not 70s, for clarity

From “was introduced in the 70s” to “was introduced in 1970”

   Line 33: Explain what is the Ellen MacArthur Foundation

Line 42-44: add “Ellen MacArthur is an organization that provides evidence-based research about CE transition and its contribution to mitigate environmental challenges such as climate change”

  Line 65: spell out VC

from “with the VCs” to “with the value chains (VC)”

     Line 75: in hypothesis: lower levels than what?  And if this hypothesis is going to be explored, needs to supported by more reference to the literature

“Lower levels” is used here to refer to rural farms and small remote producers that focus on the local market. The CE practices in these production systems are not known because the light is shed on much more important productions at the national/international levels. We defined the term of CE at lower levels.

We added a clearer definition of “lower levels:

​​” Moreover, CE is a multi-level concept, where resource circularity is distributed at two complementary separate economic structures. “High-level” circularity includes R strategies adopted by cluster industries. Whereas, “lower levels” are targeted towards actors that engage towards circularity, but on a smaller scale. The combination of CE activities at lower levels provides a higher level of circularity [6].”

 Line 77: Fuller definition of GI needed

From “One type of such food system where this hypothesis can be tested consists of geographical indications (GI). These products….”

 to “One type of such food system where this hypothesis can be tested consists of geographical indications (GI). GIs are quality schemes that are used to distinguish products, mainly from the food sector, characterized with specific quality and reputation, linked to their geographical origin. These products….”

  Line 112: Choose either ‘should’ or ‘how’ for the question

From “should/how can the Code of Practice (CoP) highlight sustainable and circular aspects, or should it remain flexible?” to “How can the Code of Practice (CoP) highlight sustainable and circular aspects, or should it remain flexible?”

   Line 113: After the 3 questions, add a sentence indicating how asking and answering these questions provides a basis for contributing to the literature

After the three research questions, we added “ ​​The findings of these questions will provide a primary understanding of how GIs appropriate CE principles in their VC and the regulatory framework that can promote and support sustainable practices. This research will have a practical implication for producers, by building a cheese VC flow chart that highlights CE and sustainability elements found in three cheese mountain products.” (line 144-148)

 Line 135: Clarify that the data was collected by other researchers, if this is the case. And clarify the role of the authors of the paper in relation to data collection and analysis. Given that no quotes are presented from the interviews, please provide some more detail on how the interviews were analyzed

We performed the interviews but the context of each value chain was based on information previously collected from the MOVING experts.

 From” This analysis is based on evidence screening of CE principles among pre-existing qualitative data, collected and analyzed by the VC experts of the project MOVING about the geographical and natural resources, the sustainability, and the resilience in their respective cheese VC”

To “This analysis is based on evidence screening of CE principles among pre-existing qualitative data about the geographical and natural resources, the sustainability, and the resilience in their respective cheese VC. This data was collected by the VC experts of the project MOVING in 2022, and further used in the current research to contextualize each of the VCs. Two co-authors had a practical implication in the data collection phase in Switzerland. They were active in reviewing the synthesis reports based on the data collection for all MOVING case studies. They were as well in close contact with the researchers of the other case studies, organizing 10 workshops with the partners of all MOVING case studies and several bilateral in-depth reviews of the cases.”

     Line 141: does this mean that the total number of participants for the paper is 8?  It is fine if yes, but need to clarify this is the case.

Line 251: from “the experts presented their VC…” to “the 8 experts in total presented…”

   Line 143: Are all the participants male? ie use of word ‘his’

Changed to “their”

   Line 146: define ‘interprofession’

From “Tête de Moine is a Swiss Alpine type PDO raw cow-milk cheese produced in the Swiss Jura region. Its strong and democratic “Interprofession'', composed of farmers, cheese makers and ripeners is responsible for managing the quality, production, and marketing of the product.”

Changed to “Tête de Moine is a Swiss Alpine type PDO raw cow-milk cheese produced in the Swiss Jura region, and managed by the interbranch organization, as defined by the Swiss. It is a strong and democratic association composed of farmers, cheese makers and ripeners, responsible for managing the quality, production, and marketing of the product. This is all defined in the Ordinance (nr. 919.117.72)

on the extension of mutual assistance measures

of interbranch organizations and producer organizations.”

   Line 158: More specifics needed on what was involved in the knowledge assessment

 From “There was also a knowledge assessment of the CE among the experts using a scale from 1, being the weakest, to 5, being the strongest.” 

to “There was also a knowledge assessment of the CE among the experts using a scale from 1, being the weakest, to 5, being the strongest. Knowledge assessment includes questions about their familiarization with the CE principles, their participation in CE projects as well as their confidence in their knowledge related to this concept.”

  Line 162: Consent is mentioned, any other ethics processes involved. Was formal ethics clearance obtained? Was permission obtained to name the organisations?

Before starting the interviews, the experts were asked for their formal consent to participate in this data collection and to record the interview. All the experts accepted for the interviews to be video-recorded.

Line 251 “Interviews were recorded, after asking for formal consent and taking the experts’ approval.”

   Line 168: how were the 5 categories of practice developed

The categories were developed to answer the research questions.

From “The qualitative data collected were analyzed following two main criteria. Firstly, the practices identified were classified per VC based on the 5Rs categories. Secondly, each practice was as well assigned to one of the following five categories: historical and still existing, historical abandoned, historical upgraded, innovative, and if present in the CoP.” 

to “ historical abandoned, historical upgraded, innovative, and if present in the CoP. This classification helps to develop a visual representation of the CE practices identified within the VC. A flow chart of the cheese processing was developed to emphasize the CE practices identified. Moreover, an applicability analysis of possible circular practices to be applied in the cheese production sector was elaborated as a guide of good practices for producers.

Line 167: more clarity is needed here ie restate that this is based on 8 experts, if that is the case.

Directly below figure 2, we added: “ In total, 8 experts participated in the interviews”

Line 192: The Table is good. However, with each line of the table, is it possible to identify how many of the 3 VCs were using the practice?

Yes, thank you for this suggestion. We updated the Table (1).

Line 351: Figure 3 is an important contribution

Thank you for the validation of our work.

    Line 360: A scale of 5 + for cost is mentioned. How was it decided where a particular practice fit in terms of the scale?  It is mentioned that they are estimated and summarised. What is the basis for the estimate?

During the interviews, the experts would highlight information about the cost-efficiency of the technology used within their VC. This helped us develop an estimate of the cost of each of the practices found in this table.

From: “The cost, expressed by an ascending 5 scale “+” sign, and the benefits are estimated and summarized based on the discussions that took place with the experts of the VC. “

To: “The cost, expressed by an ascending 5 scale “+” sign, was estimated based on the discussions from the three interviews with the experts about the cost-efficiency of the practices or technologies. The benefits are as well estimated and summarized from the information collected during the interviews that took place with the experts of the VC.”

      Line 362: Another sentence is needed to explain how Table 4 will help producers rethink their systems. While the Table contains important information, it is not clear how it will be operationalised by producers

From “table 4 can act as a guide that will help producers to rethink their production system and guide them in putting in place possible circular practices”

to “Table 4 can act as a guide that will help producers to rethink their production system and guide them in putting in place possible circular practices. Such evidence-based guide represents successful CE practices or technologies adopted, aimed at possible environ-mental targets found in the cheese production system. This as well helps producers understand the short and long-term benefits, economically and environmentally, of implementing these practices in their production system. “

   Line 393: It is claimed that public actors and local authorities emerge as the key actors. This might be the case, but what evidence is the claim based on?

Thank you for this comment.

From “Public actors and local authorities emerge as the key actors to provide the constitutional and structural framework for transitioning towards a CE….”

 to “Previous work on agriculture research impact (Barjolle et al., 2018) show that if a constitutional and structural framework for agriculture research aims at reaching out towards the local actors, the change and adoption of new approaches and technologies increases. This was as well suggested by the evidences gathered among all cases studies, that most of the respondents were interested to get support from research and outreach to be more informed about the circular economy principles and related technologies”.

 Line 455: I would suggest not introducing other examples at this point.

Thank you for this suggestion. We removed the example about Champagne introduced at the discussion part of the article.

Towards the end of the paper, return to the central questions and provide answers. Also indicate to what extent the hypothesis has been supported.

Indeed, referring to the main research questions in the discussion is important to provide answers.

Round 2

Reviewer 1 Report

The authors have revised the paper.